# LFA-1 Activation in T-Cell Migration and Immunological Synapse Formation

**DOI:** 10.3390/cells12081136

**Published:** 2023-04-12

**Authors:** Huiping Shi, Bojing Shao

**Affiliations:** 1Cardiovascular Biology Research Program, Oklahoma Medical Research Foundation, Oklahoma City, OK 73104, USA; 2Department of Biochemistry and Molecular Biology, University of Oklahoma Health Sciences Center, Oklahoma City, OK 73104, USA

**Keywords:** LFA-1, T-cell migration, immunological synapse

## Abstract

Integrin LFA-1 plays a critical role in T-cell migration and in the formation of immunological synapses. LFA-1 functions through interacting with its ligands with differing affinities: low, intermediate, and high. Most prior research has studied how LFA-1 in the high-affinity state regulates the trafficking and functions of T cells. LFA-1 is also presented in the intermediate-affinity state on T cells, however, the signaling to activate LFA-1 to the intermediate-affinity state and the role of LFA-1 in this affinity state both remain largely elusive. This review briefly summarizes the activation and roles of LFA-1 with varied ligand-binding affinities in the regulation of T-cell migration and immunological synapse formation.

## 1. Introduction

Timely and proper interactions with the vascular wall and other immune cells are critical for T cells to exert immune surveillance functions and fight against invading pathogens or neoplasms. LFA-1 (lymphocyte function-associated antigen 1, αLβ2, and CD18/CD11a) is a key integrin on T cells that plays an essential role in the regulation of these interactions. To mediate these interactions of T cells, upregulating the ligand-binding affinity of LFA-1, termed LFA-1 activation, is a prerequisite cellular event, because it allows ligands to bind LFA-1. In T cells, LFA-1 is activated by signals that are transmitted from inside of the cells to the outside; generally called inside-out signaling. Ligand binding to activated LFA-1 also induces signals, which take the reverse direction and are called outside-in signaling. This bi-directional signaling enables T cells to respond rapidly to changes of both intracellular and extracellular environment. Furthermore, LFA-1 is presented with varied ligand-binding affinities on T cells in differing functional states. Thus, by interacting with ligands and mediating signaling, LFA-1 regulates T-cell migration into lymphoid organs or afflicted sites where T cells mature, differentiate, or have effector functions. Consistent with its diversified roles, dysregulated ligand binding to LFA-1 on T cells causes many inflammatory and autoimmune conditions, such as viral infection, arthritis, diabetes, inflammatory bowel disease, and psoriasis. Accordingly, the temporal and spatial regulation is never an overstated aspect in studying LFA-1 on T cells in the host immune responses. Here we will review LFA-1 activation in T-cell migration during inflammation and immune responses in immunological synapses.

## 2. Conformations of LFA-1 on T Cells

Distinct ligand binding abilities (affinities) are controlled through regulating conformations of the LFA-1 extracellular domain [1]. The extracellular domain of LFA-1 can be divided into ‘head’ and ‘leg’ regions (Figure 1). The β-propeller domain of the αL subunit and the β I domain of the β2 subunit form a ‘head’ at the N-terminal region of the extracellular domain [2,3,4]. The α I domain, which is the major ligand-binding site, is inserted into the β-propeller. The regions from the C-terminal to the β-propeller domain and the β I domain comprise the ‘leg’ of the extracellular domain. The ‘leg’ of the αL subunit contains the thigh domain, the genu, and the calf-1 and -2 domains. The upper β ‘leg’ has the hybrid domain and the plexin-semaphorin-integrin (PSI) domain. The β I domain is inserted in the hybrid domain, which in turn is inserted in the PSI domain. The lower β ‘leg’ contains four integrin epidermal growth factor-like (I-EGF) domains and a β tail domain.

Structurally, the integrin ‘head’ can be opened when the hybrid domain swings away from the αL subunit. Interactions of the α7 helix in the β I domain with the membrane proximal β tail domain bend the ‘leg’ at the ‘knee’, which locates at αL subunit genu and between I-EGF1 and I-EGF2 domains of the β2 subunit. Movements in ‘head’ and ‘leg’ regions allow the extracellular domain of integrin LFA-1 to adopt at least three distinct conformations: bent ‘leg’ with a closed ‘head’ piece, extended ‘leg’ with a closed ‘head’ piece, and extended ‘leg’ with an open ‘head’ piece. Differing conformations of the LFA-1 extracellular domain correspond to distinct ligand-binding affinities. A closed and an open ‘head’ piece on the extended extracellular domain allow LFA-1 to bind its ligands with the intermediate and high affinities, respectively [5]. LFA-1 with a bent extracellular domain and a closed ‘head’ exhibits a low affinity for ligands [5]. Compared to the bent-close conformation, LFA-1 with the extended-open conformation has a 10,000-fold higher affinity for ligands [6]. In addition, β2 integrins on neutrophils with a bent ‘leg’ can have an open ‘head’ and bind to intercellular adhesion molecule (ICAM)-1 or ICAM-3 with a high affinity through *cis* interactions [7]. However, neutrophils express several β2 integrins, such as LFA-1 and Mac-1 (αMβ2, CD18/CD11b); it is not clear whether LFA-1 can have an open ‘head’ on the bent ‘leg’. LFA-1 is the only β2 integrin on T cells. LFA-1 on T cells can be presented with a bent ‘leg’ with a closed ‘head’ and an extended ‘leg’ with a closed or an open ‘head’ [8,9,10,11]. Beside three principal conformations, transitional conformations of integrin β2 also exist.

## 3. Signaling Pathways to Activate LFA-1 on T Cells

The ligand-binding affinity of LFA-1 on T cells is mainly regulated through inside-out signaling, in which talin1 binding to the cytoplasmic domain of the β subunit is the final step to activate integrins [12,13,14]. Talin1 is a large adaptor protein consisting of an N-terminal FERM (band 4.1, ezrin, radixin, and moesin) domain (head) and a C-terminal rod domain [15,16]. The talin1 head is essential and sufficient to activate integrins. Talin1 engages integrins via the interaction of phosphotyrosine-binding (PTB) domain on F3 subdomain in the talin1 head with NPxY/F (x, any amino acid) motif in the tail of integrin β subunit [16,17]. The integrin β subunit has two tandem NPxY/F motives. The talin1 head domain binds with high affinity to a membrane-distal NPxY/F motif on the β2 tail, which facilitates binding to a low-affinity, membrane-proximal NPxY/F site on the β2 tail [18,19]. Binding of talin1 to the NPxY/F motif proximal to the cell membrane disrupts a salt bridge between integrin α and β tails [20], tilts the crossing angle of the β integrin transmembrane domain [21], and extends electrostatic interaction of talin1 with the phospholipid head of cell membrane [22,23]. These events consequently trigger conformational changes of the integrin extracellular domain (i.e., increasing the ligand-binding affinity) that leads to integrin activation.

On quiescent T cells LFA-1 is in a low-affinity state, and LFA-1 is activated through the signaling triggered by engagement of T-cell receptors (TCRs), chemokines, or selectins. The ligation of TCRs by peptides presented with major histocompatibility complex (MHC) molecules and the binding of chemokines to their receptors trigger talin1 binding to the NPxY/F motif that converts LFA-1 from the low- to the high-affinity state (Figure 2) [1,7,8,24,25,26]. Upon engagement of P- and E-selectin, talin1 binds the NPxY/F motif that upregulates LFA-1 to the intermediate-affinity state (Figure 2) [11,27]. P- and E-selectins are transmembrane Ca^2+^-dependent lectins that mediate leukocyte rolling on the inflamed vessel wall. Ligand binding to selectins induces tyrosine phosphorylation of multiple proteins in leukocytes [28]. Modulating the conformation of integrin LFA-1 in neutrophils is the extensively studied physiological relevance of selectin signaling. The selectin-mediated signaling in activating LFA-1 also exists in effector T cells, but not in naive T cells [11], as naive T cells lack the minimal recognition determinant for selectins, sialyl Lewis x on the terminus of some O- and N-glycans with sulfates, due to defects of glycosylation or sulfation [29]. LFA-1 in the intermediate- and high-affinities are distinct from each other in the activation approach and in the consequent signaling upon ligand binding. As an example, inhibiting chemokine signaling with pertussis toxin does not impair activation of LFA-1 by selectin engagement [11,27,30]. Mutation of *talin1* (L325R) abolishes the interaction with the membrane proximal NPxY/F motif on integrin β3 tail [18], and prohibits activation of platelet integrin αIIbβ3 to the high-affinity state [31]. This mutation also prevents activation of LFA-1 to the high-affinity state by chemokines, but spares selectin-induced activation to the intermediate-affinity state [32]. Thus, LFA-1 activation to differing affinity states requires talin1 binding to varied sites on the β2 tail. In addition, kindlin3 is required for LFA-1 activation to extended-open conformation, but not to extended-closed conformation [33]. Phosphorylation of tyrosine 145 and tyrosines 112/128 in adaptor SLP-76 is triggered by engagement of chemokines and P-selectin during LFA-1 activation, respectively [34,35].

The most significant difference between these two affinity states of LFA-1 may arise from the differing reliance of their activation on the association with actin cytoskeleton. The cytoplasmic domain of integrin β tail is associated with actin cytoskeleton via adaptors such as talin1 and kindlins [36]. The cytoskeleton forms a scaffold for the signaling complex and serves as a transaction point converting mechanical force in the extracellular environment into biochemical signals inside cells. The cytoskeleton association with LFA-1 further separates the integrin α and β chains and facilitates activation of LFA-1 to the high-affinity state, extended-open conformation [37,38,39,40]. Tensile force exserted by the actin cytoskeleton also orientates LFA-1 on the plasma membrane and stabilizes LFA-1 conformation in the high-affinity state [37,41]. In addition, association with the cytoskeleton also induces the formation of integrin clusters [42]. Unlike upregulating the ligand-binding affinity of individual integrin upon activation, integrin clustering increases the number of integrin-ligand bonds (i.e., valency). Both integrin affinity and valency determine the total strength of the integrin-ligand interaction (i.e., avidity). Furthermore, upon ligand binding to LFA-1 in the high-affinity state, integrins rapidly transduce signaling to alter cell morphology and enhance cell-cell contact and cellular responses [42]. LFA-1-mediated signaling also requires association with the actin cytoskeleton [43,44]. In sharp contrast, intact actin filaments or actomyosin tension are dispensable for LFA-1 activation to the intermediate-affinity, extended-closed conformation [45]. Kindlin3 plays a critical role in the formation of integrin clusters [46] but is not required for selectin-induced LFA-1 activation [33]. These findings seem to suggest that the intermediate-affinity LFA-1 does not associate with the cytoskeleton. However, in chemokine-stimulated T cells, LFA-1 in the intermediate-affinity state associates with the actin cytoskeleton through binding α-actinin-1 [8,47]. Given selectin signaling is transient and leukocytes keep rolling without adhesion, whether the actin cytoskeleton associates with LFA-1 in the intermediate-affinity state in response to selectin engagement remains elusive. If so, do actin cytoskeletons play a role in the signaling of LFA-1 with the intermediate affinity in T cells rolling on selectins, on which duration for ligand binding is short? Addressing this interesting question would enhance our understanding of the mechanisms for LFA-1 activation.

LFA-1 activation through differing pathways may be distinguished from varied effects on phosphorylation of integrin β2 tail. The cytoplasmic domains of both integrin α and β subunits can be phosphorylated [48]. Phosphorylation of integrin β2 tail can either promote or inhibit LFA-1 activation. For example, docking protein 1 (Dok1) binds phosphorylated Ser756 (pSer756) on integrin β2 [49], thereby competing with talin1 for integrin binding and inhibiting integrin activation [50]. 14-3-3 docks to phosphorylated Thr758 (pThr758) leading to reorganization of the actin cytoskeleton and increased T-cell adhesion [51,52]. Signaling of chemokines and TCR triggers de-phosphorylation of Ser756 on integrin β2 [49], as well as activating protein kinase C to phosphorylate Thr758 [51,53,54]. Whether selectin signaling alters phosphorylation of integrin β2 tail and whether LFA-1 in the intermediate-affinity state requires pThr758 and/or Ser756 have not currently been studied. As selectin signaling does not induce morphology changes of leukocytes, a subsequent cellular response of actin reorganization, selectin-mediated LFA-1 may not alter phosphorylation of β2 tail. If so, LFA-1 in the intermediate affinity upon engagement of selectins and chemokines would have differing functions in cell migration and immune functions. 

LFA-1 activation also exhibits differential requirement for immunoreceptor tyrosine-based activation motif (ITAM)-containing adaptors. Selectin signaling resembles signaling of immunoreceptors such as TCR, which activates tyrosine kinases sequentially and recruits adaptors to propagate signaling. The pathway of selectin signaling in neutrophils requires ITAM-bearing adaptors, DAP12 and FcRγ [55]. ITAM motif is a tandem sequence, YxxL/Ix6-8YxxL/I (x, any amino acid), in the cytoplasmic region of various non-catalytic tyrosine-phosphorylated transmembrane receptor proteins. ITAM motif is a critical mediator of intracellular signals. Phosphorylation of tyrosine residues in ITAM motif serves as docking sites to recruit Syk/ZAP70 propagating cellular signaling to the downstream [56,57]. DAP12 and FcRγ chain are the ITAM-bearing proteins in myeloid cells, while TCRζ chain, and CD3 δ, ε, γ subunits are the predominant ITAM-containing receptors in T cells [56,57]. Recently, we have demonstrated that DAP12 is also expressed in type1 T helper cells (Th1 cells) [11], increasing the approaches to regulate intracellular signaling in T cells. Whether DAP12 is expressed in other types of T cells remains to be examined. ITAM-bearing receptors in T cells, such as CD3 subunits, are essential for LFA-1 activation to the high-affinity state by signals from engagement of TCR, but not chemokines [11,58]. DAP12 is required for selectin-induced activation of the intermediate-affinity state LFA-1 [11,55]. DAP12 is also involved in signaling of ligand-binding to LFA-1 [11]. Although ITAM-bearing adaptors are essential in LFA-1 activation by TCR and selectin, whether DAP12 has synergistical effects with CD3 subunits on LFA-1 activation remains elusive.

## 4. Regulation of T-Cell Migration by LFA-1

The most remarkable feature of immune cells is their ability to travel throughout the body and cross boundaries to protect their hosts from infection and to maintain tissue homeostasis. Extravasation of T cells from circulation into inflamed or injured tissues and lymph nodes is essential for T-cell function and maturation [59]. This cell trafficking is a highly organized multistep cascade, including tethering/rolling, arrest, firm adhesion and spreading, crawling, and transmigration between or through the endothelium into tissues [60]. Tethering or rolling of T cells on the endothelium is initiated by P- and E-selectin in inflamed tissues [61,62] and peripheral node addressins (PNAd) in lymph nodes [63]. Tethering or rolling on the endothelium facilitates the interaction of chemokines with their receptors, which induces inside-out signaling to activate integrins to the high-affinity state leading to T-cell arrest. Several integrins including LFA-1, VLA4 (α4β1, CD49d/CD29), and LPAM-1 (α4β7, CD49d/β7) are involved in T-cell arrest [64,65,66,67], and differing integrins show tissue specificity in T-cell trafficking during inflammation. For example, LFA-1 plays a major role in migration of T cells in a bronchial epithelial model [68], whereas LPAM-1 is essential in regulating T-cell recruitment to skin in a model of contact hypersensitivity [69]. On arrested T cells, ligand binding to integrins in the high-affinity state triggers outside-in signaling, which causes reorganization of the cytoskeleton to strengthen cell adhesion (i.e., firm adhesion) [60] and to induce elongation of T cells (i.e., spreading and uropod formation) [70]. After adhesion, integrins also guide T-cell migration through (trans-endothelial migration) or between (para-endothelial migration) endothelial cells into underlying tissues. T cells form specialized podosomes; these are protrusions with LFA-1 and talin-1 in outer rings and rich F-actin in inner cores. Interactions of LFA-1 in podosomes with ICAM-1-enriched invaginations, called podoprints, on the endothelium form a transcellular pore, thus leading to transcellular migration of T cells [71]. LFA-1 also binds junctional adhesion molecule-A (JAM-A) at the apical side and tight junction of endothelial cells to control the direction of T-cell para-endothelial diapedesis [72]. In addition, as a mechanical sensor, shear force applied on the high-affinity LFA-1 also increases the density of filopodia, the protrusion in the front of crawling cells, to search for sites of extravasation [9]. However, as the role of VLA-4 and LPAM-1 in regulating the directional migration of T cells remains elusive [73], more studies are required to ascertain whether LFA-1 still guides T-cell extravasation in tissues in which VLA-4 and LPAM-1 dominate cell trafficking.

During effector T-cell trafficking into inflamed tissues, LFA-1 also presents an intermediate-affinity state [11]. Unlike T-cell integrins in the high-affinity state, which are activated by the signaling of chemokines and TCR and arrest leukocytes onto the endothelium [64,74], LFA-1 with the intermediate-affinity is induced by P- and E-selectin engagement. With the intermediate affinity, LFA-1 forms reversible interactions with its ligands, such as ICAM-1. The rapid formation and breakage of integrin-ligand interactions slow the rolling velocities of leukocytes by ~50%, but these interactions do not mediate cell adhesion [11,27]. Slowing rolling velocities by LFA-1 in the intermediate-affinity increases encounters of chemokines with their receptors, thus facilitating leukocyte adhesion and migration [75]. Alternatively, selectin signaling may prime leukocytes to respond to suboptimal levels of chemokines [75]. Blocking selectin signaling by deleting key signaling proteins has a minimal effect on recruitment of T cells and neutrophils in several models of inflammation [11,27,30]. In addition, disabling chemokine receptors by pertussis toxin only partially inhibits leukocyte recruitment [11,27]. Thus, selectins and chemokines cooperate to maximize leukocyte recruitment, in terms of LFA-1 activation. Furthermore, LFA-1 in those two affinity states may also play differing roles in regulating leukocyte migration. Studies indicate that activated LFA-1 has distinct locations on the migrating T cells. The resting leukocytes have two surface domains: flat surface and microvilli. LFA-1 is on the flat surface. On trafficking leukocytes, signaling from chemokines and ligand-bound integrins activates protein kinase C and Rho family G-proteins Cdc42 and Rac, leading to activation of myosin light chain kinase and polymerization of actin [42,76]. Activated myosin then contracts the cortical actin cytoskeleton which leads to collapse of the cortical actin cytoskeleton to the side of cells. Consequently, myosin retraction causes new polymerization of the actin resulting in polarization of migrating leukocytes. This rearrangement of the actin cytoskeleton in polarized cells forms membrane protrusions at the leading edge. Meanwhile, motile leukocytes have the main cell body and the rear edge (posterior uropod) in morphology. Leukocyte polarization is functionally important. For example, the leading edge is highly motile that increases interactions with the extracellular environment. In addition, forces generated from the actin polymerization enhance leukocyte adhesion [77]. The rear edge may facilitate detachment of cell adhesion promoting cell directional movement. Corresponding to changes of cell morphology, membrane proteins may also exhibit altered distributions. For integrins on migrating T cells, most LFA-1 molecules are in the high-affinity state and locate in the mid-cell focal zone (i.e., main cell body). In actin-based protrusions at the leading edge in the front, a lower level of LFA-1 in the high- and intermediate-affinity states distributes in filipodia (thin spikes generated by activated Cdc42) and lamellipodia (flat “ruffles” generated by Rac), respectively [8,9]. LFA-1 in the focal zone controls adhesion of T cells, while LFA-1 in the leading edge regulates the direction and speed of extravasation of T cells [8,9]. To be noted, on trafficking leukocytes selectin receptors locate in the rear uropod and protrude into the lumine of the blood vessel [78,79]. Yet, selectins seem to transduce short-distance local signals to activate LFA-1. Therefore, in polarized motile cells whether LFA-1 in the intermediate-affinity state is activated by selectin signaling in a long distance and whether selectins still cooperate with chemokines in regulating leukocyte transmigration remains to be studied.

LFA-1 also orchestrates with integrin VLA-4 (α4β1) in regulating T-cell migration. The interaction of LFA-1 with ICAM-1 supports T-cell adhesion and upstream crawling with the constant speed, while the interaction of VLA-4 with its ligand, vascular cell adhesion molecule-1 (VACM-1), mediates transient adhesion and downstream crawling with the faster speed [80]. Activation signaling for LFA-1 and VLA-4 cross-talks to each other. For example, pThr758 in integrin β2 causes dephosphorylation of Thr788/789 in integrin β1, resulting in inactivation of VLA-4 [81]. This inhibition signaling may be through the 14-3-3/Rac-1 route. Yet, VCAM-1 binding to VLA-4 stimulates LFA-1-mediated migration through activating focal adhesion kinase and proline-rich tyrosine kinase-2 [82]. In addition, LFA-1 and VAL-4 in high-affinities locate in the front leading edge and rear body, respectively [80]. This polarized distribution of integrins may facilitate detachment of migrating cells from the blood vessel wall as well as the directional extravasation. As both VLA-4 and selectins mediate leukocyte rolling and stimulate LFA-1 activation, whether those two adhesion molecules have synergic effects on T-cell adhesion is an interesting question. 

## 5. LFA-1 Functions in Immunological Synapse Formation

Immunological synapse (IS) is a communication zone formed when T cells interact with antigen-presenting cells (APCs). IS serves as a platform in regulating antigen recognition as well as T-cell activation, differentiation, and memory formation, which is critical for hosts to fight against virus infection and cancer. The formation of IS also is a multistep process that begins with adhesion of T cells to APCs. LFA-1 initiates the interaction of T cells and APCs as well as regulates T-cell activation, thus playing a critical role in the IS formation. Once entering lymph nodes, T cells migrate rapidly to search for cognate peptide-MHC (pMHC) complex on APCs and engage APCs via transient contacts (kinapses) [83]. However, negatively charged glycocalyx on the cell membrane forms a barrier to prevent stable cell-cell contact [84]. The solution to this conflict is that activated LFA-1 interacts with its ligands such as ICAM-1 at ~40 nm in distance, thereby bringing T cells close to APCs [85]. During IS formation, T cells receive signals from chemokines, such as CCL19 and CCL21 [86], and TCR [86,87,88]. The inside-out signaling from chemokines and TCR triggers talin1 binding to the β2 tail, leading to LFA-1 activation. LFA-1 engagement not only overcomes the spatial barrier between T cells and APCs but also increases TCR-enriched active protrusions in the adjacent regions of the T-cell surface [89]. Interaction of LFA-1 with its ligand, ICAM-1, also increases the sensitivity of T cells to cognate antigens [90]. For example, TCRs in the active membrane protrusions are more sensitive to MHC-peptide complexes on APCs [89]. Thus, ligand binding to activated LFA-1 initiates and stabilizes the interface between T cells and APCs, thus ensuring adequate TCR-MHC interaction [91,92]. Upon engaging antigens, TCR also triggers signals to stabilize cell adhesion by enhancing LFA-1 activity [93], and cause rapid actin polymerization driving the TCR cluster formation as well [94]. In the contact interface, TCRs and protein-sorting and secretory compartments are accumulated in the core region, which is surrounded by an LFA-1-rich intermediate ring and a peripheral actin-rich ring [95]. T cells integrate short-lived TCR signaling until reaching a critical activation that stops T-cell migration [87]. Then prolonged TCR-specific ISs are generated. In the canonical structure of IS, TCR and associated signaling molecules, such as CD28, locate in a central region, which is surrounded by an outer ring of LFA-1 and talin1 [95,96]. These regions are named central and peripheral supramolecular activation clusters (cSMAC and pSMAC), respectively (Figure 3). The edge of IS is a distal SMAC region (dSMAC) which contains CD45 and F-actin [97]. Furthermore, the outside-in signaling from ligand binding to LFA-1 synergizes with TCR signaling to regulate actin remodeling and T-cell activation promoting IS formation and effector functions of T cells [98,99]. For example, LFA-1 signaling enhances recruitment of TCR/pMHC complex into cSMAC and segregation of the phosphatase CD45 from the immune synapse [92]. LFA-1 also tunes interactions of T cells with APCs via altering mechanical forces. Mechanical force of LFA-1 enhances T-cell engagement with pMHC on substrates, thus potentiating antigen-dependent T-cell activation and the discriminatory power of TCR against near cognate antigens [100]. Therefore, TCR signaling and LFA-1 activation enhance the signal magnitude of each other in the immune synapse and ensure proper T-cell activation.

Interestingly, LFA-1 and TCR exhibit cooperation and independence in aspects of cell adhesion and signaling during the IS formation. Both LFA-1 and TCR mediate signals to regulate remodeling of the actin cytoskeleton and tune T-cell activation. However, LFA-1 and TCR distribute in differing regions of IS. The engaged TCR moves into the center of the synapse, whereas LFA-1, even in the absence of ICAM-1 binding, remains in a discrete peripheral ring of the synapse or adhesion. The cooperation of LFA-1 and TCR in a large-scale spatial distance is always an intriguing topic to study. In addition, how to maintain LFA-1 in the peripheral ring of the synapse and the biological significance of this distribution remain elusive. It is postulated that talin1 binding may prevent LFA-1 moving to the center of IS. Talin1 is accumulated in areas of LFA-1/ICAM-1 binding in the synapse [96], and regulates immobilization of integrin complex [101]. However, it is difficult to test this hypothesis, because talin1 is required for LFA-1 to interact with its ligands. In addition, LFA-1 in the synapse is in two differing activation states, which display distinct patterns of organization at IS. The intermediate-affinity LFA-1 is enriched in a ring roughly corresponding to the pSMAC region, whereas LFA-1 in the high-affinity for ligands is concentrated in a more central ring [38]. Studies of bond lifetimes suggest that LFA-1 in the high-affinity state at IS interacts with ICAM-1 [102]. This prompts the interesting question of whether LFA-1 in the intermediate-affinity state at IS has biological functions. In addition, studies with neutrophils suggest that LFA-1 activation to the distinct activation states is due to talin1 binding to differing NPxY motifs on the β2 cytoplasmic domain with chemokine and selectin signaling [32]. How to regulate and maintain activation of LFA-1 to distinct ligand-binding affinities during the IS formation is unknown. Addressing this question may provide insights into fine tuning of the formation and functions of IS.

Besides IS between T cells and APCs, cytotoxic T cells can form cytotoxic IS (cIS) with tumor cells [103]. In cIS, LFA-1 is activated by TCR signaling and is located at the periphery region, which is similar to the pSMAC region in IS. ICAM-1 engagement to LFA-1 in the high-affinity state triggers T cells to release cytokines and lytic granules, thereby killing tumor cells. In addition, LFA-1-mediated adhesion increases the junction area of cIS and directs the release of cytotoxic granule contents into a restricted ring-like region [74,104], which stabilizes cIS and mediates the effective destruction of targeted tumor cells. These findings are evidenced by reduced T-cell cytotoxicity with blocking antibodies to LFA-1, M7/14 [105]. Chimeric antigen receptor T (CAR-T) cells are genetically engineered cytotoxic T cells that effectively kill tumor cells [106]. CAR-T cells form non-classical cIS with tumor cells [107,108]. In cIS of CAR-T cells LFA-1 distributes diffusively and distinct LFA-1 adhesion rings lack, which causes a faster off-rate from tumor cells [108]. Compared with cIS of native cytotoxic T cells, CAR-T cIS initiates shorter but more rapid signaling [108]. Thus, CAR-T cells kill tumor cells and detach from dying cells more rapidly. In contrast, native cytotoxic T cells may kill tumor cells more extensively. To be noted, LFA-1 in IS may play a role in T-cell suppression [109]. Recent studies show that LFA-1-mediated adhesion promotes interactions of programed death ligand 1 (PD-L1) on tumor exosomes with PD-1 on T cells leading to inhibition of T cells. Clearly, the role of LFA-1 in cIS for cancer therapy may vary within differing cellular milieu. In addition, whether LFA-1 exists with the intermediate-affinity state in cIS and the benefit of manipulating the ligand-binding affinity of LFA-1 in cIS both need to be determined.

## 6. Summary and Future Outlook

LFA-1 is a major integrin that plays a critical role in T-cell migration and the formation of IS through interacting with its ligands. The ligand binding affinity of LFA-1 is controlled by intracellular signaling and corresponds with the conformation of the extracellular domain of LFA-1. LFA-1 on T cells exhibits three distinct affinity states for ligands: low-, intermediate-, and high-affinities. Most studies focus on the role of LFA-1 in the high-affinity state in regulating T-cell functions. LFA-1 on effector T cells can be activated to the intermediate-affinity state by selectin engagement [11]. LFA-1 in this affinity state is also presented in the leading edge of migrating T cells [8] and the pSMAC region of IS [38]. However, how to regulate LFA-1 activation to the intermediate-affinity state in IS and the leading edge of T cells as well as the biological significance of LFA-1 in this affinity state have not been well studied. 

The aberrant activation of LFA-1 on T cells contributes to the development of infectious diseases, cancer, and autoimmune diseases. Thus, LFA-1 has been a target to modulate T-cell functions to fight infection or induce self-tolerance for autoimmunity. However, in contrast to the success of inhibitors against integrins αIIbβ3, α4β1, and α4β7, none of the LFA-1 targeted drugs, either antibodies, peptides, or small molecules are clinically successful [110]. The major reason is the broad effects of LFA-1 signaling in regulating immunological responses [111]. The intermediate-affinity LFA-1 maintains the minimum cell-cell interactions and may have differing signaling than the high-affinity LFA-1, therefore, further identifying the differential regulation of LFA-1 activation in T cells may help develop a novel approach to regulate T-cell functions.

Targeting LFA-1-related adaptor proteins may also provide a novel approach to tune the effects of LFA-1 activation differentially. As an example, RASA3, a Ras/Rap GTPase-activating protein, negatively regulates LFA-1-mediated T-cell adhesion and activation [112]. Blocking RASA3 might modulate LFA-1 function and T-cell activity in autoimmune diseases. As a new ITAM-bearing adaptor, the role of DAP12 in T-cell activation and functions has not been studied. Human patients with loss-of-function mutations in the DAP12-encoding *TYROBP* gene mainly displayed presenile dementia and bone cysts [113]. DAP12-deficient mice showed significantly reduced IFN-gamma production by myelin-reactive CD4^+^ T cells and inadequate T-cell priming [114]. As an adaptor involved in LFA-1 activation and signaling, the role of DAP12 in regulating LFA-1-mediated T-cell functions would be interesting to examine.

Immunotherapy provides substantial benefits for some cancer patients, yet the treatment efficacy in most tumor types remains limited [115]. One key reason is the immunosuppressive tumor microenvironment, which reduces recruitment of cytotoxic T cells. Integrins including LFA-1 regulate T-cell recruitment. Recent studies show that a small-molecule activator of LFA-1, 7HP349, enhances tumor killing activity by increasing recruitment of tumor-specific T cells into the tumor microenvironment [116]. However, LFA-1 mediates intra-tumor activated T cells to form clusters in mouse models; this reduces egress of T cells from the tumor and suppress immune responses in the draining lymph nodes [117]. Thus, adjusting strategies according to the pathogenesis of differing tumors may improve the efficacy of cancer therapy.

Taken together, we summarized the critical role of integrin LFA-1 with differing ligand-binding affinities in T-cell migration and IS formation. Ligand-binding affinities that correspond with conformations of the extracellular domain of LFA-1 are regulated through multiple signaling pathways and exhibit varied effects during T-cell migration and formation of IS. Although extensively studied, much remains unknown in the regulation and functions of LFA-1, especially for LFA-1 in the intermediate-affinity state. Further understanding the role of LFA-1 presenting different conformations in T cells is of great significance as LFA-1 is an important regulator of T-cell biology. Clinically, LFA-1 modulation may serve as an effective method to treat human disorders including but not limited to infections, autoimmune diseases, and cancers.

## Figures and Tables

**Figure 1 cells-12-01136-f001:**
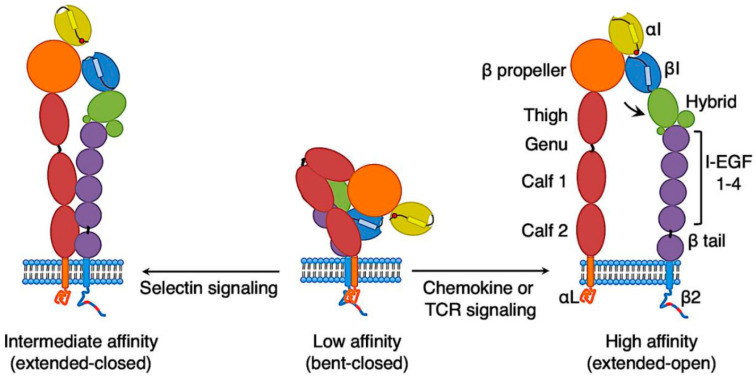
Distinct LFA-1 conformations and affinities in T cells. The low-affinity LFA-1 adopts the conformation of a bent ‘leg’ with a closed ‘head’. Inside-out activation of LFA-1 by signaling from T-cell receptor (TCR) or chemokines triggers its conformational change to a high-affinity state: an extended ‘leg’ with an open ‘head’. In contrast, selectin engagement leads to intermediate-affinity state LFA-1 in effector T cells: an extended ‘leg’ with a closed ‘head’.

**Figure 2 cells-12-01136-f002:**
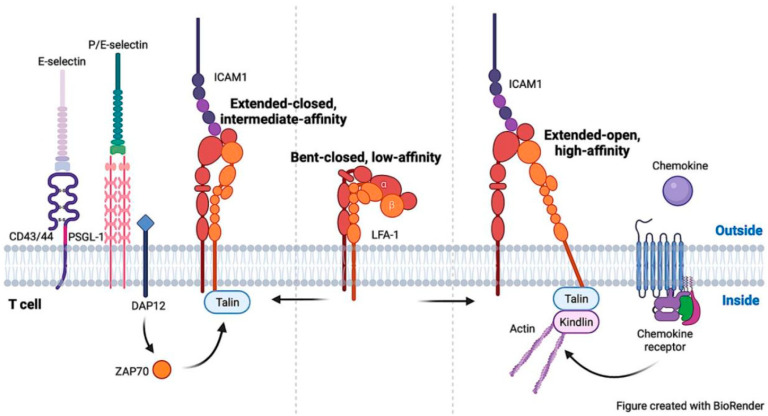
LFA-1 activation during T-cell migration, conformations and affinities in T cells. Signaling from engagement of selectins and chemokines during T-cell migration activates LFA-1 to the intermediate- and high-affinity states, respectively. Talin1 binding to different sites on the integrin β2 tail of the intermediate- and high-affinity LFA-1. LFA-1 in the high-affinity state associates with the actin cytoskeleton, but it is not clear whether the intermediate-affinity LFA-1 is linked to the cytoskeleton in response to selectin engagement. In addition, kindlin3 is required for LFA-1 activation by chemokines but not selectins. Interestingly, DAP12 is expressed in effector T cells and plays an essential role in selectin-mediated LFA-1 activation.

**Figure 3 cells-12-01136-f003:**
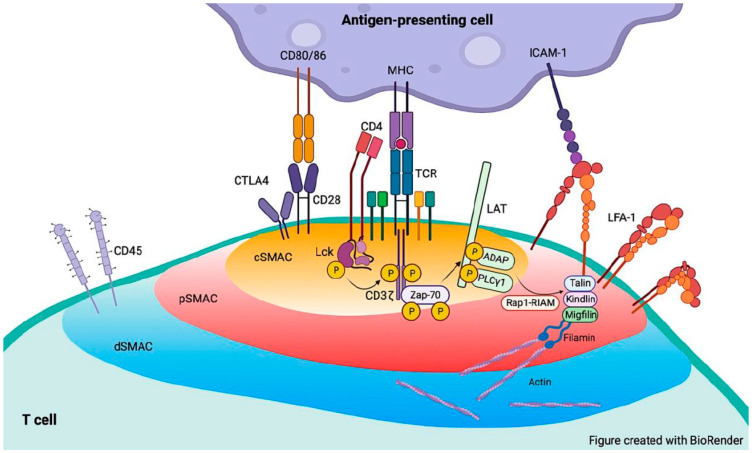
Role of LFA-1 in the immune synapse formed between antigen-presenting cells and T cells. The immune synapse is characterized by different compartments: central, peripheral, and distal supramolecular activation clusters. T-cell receptors (TCR) and associated signaling molecules are in the central supramolecular activation cluster (cSMAC). The peripheral supramolecular activation cluster (pSMAC) is an outer ring rich in LFA-1 and talin1. The distal supramolecular activation cluster (dSMAC) contains CD45 and F-actin. LFA-1, leukocyte function-associated antigen 1; ICAM-1, intercellular adhesion molecule 1; CTLA4, cytotoxic T lymphocyte antigen 4; Zap-70, ζ-chain-associated protein 70; LAT, linker for activation of T cells; MHC, major histocompatibility complex; Lck, lymphocyte-specific protein tyrosine kinase; ADAP, adapter protein; PLCγ1, phospholipase Cγ1; Rap1, Ras-related protein 1; RIAM, Rap1-GTP-interacting adaptor molecule.

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
