# Peer review of "LFA-1 Activation in T-Cell Migration and Immunological Synapse Formation"

_cells, 2023, doi:10.3390/cells12081136_

Round 1
Reviewer 1 Report
This is a thorough review of the role of LFA-1 in T cell biology with a focus on migration and cellular interactions on a molecular level. I did not find any inconsistencies. However, the manuscript might benefit if a sub-chapter extends the discussion to broader functional phenomena such as T cell plasticity, T cell cytotoxicity, T cell exhaustion, effector-memory conversion etc. This would give the paper a perspective towards clinical application, such a checkpoint inhibitor therapy, and also attract readers from other fields that have not come in contact with LFA-1 biology before.
Reviewer 2 Report
The review ”LFA-1 activation in T cell migration and immunological synapse formation” provides an overview of the structural regulation of the LFA-1 integrin and how this affects its function as an adhesion molecule regulating migration processes and its role in the immunological synapse. Focus is on the intermediate form of the integrin, which distinguishes it from several other recent integrin reviews. The review is comprehensive and easy to follow and the English language for the most part good, with minor spelling errors.
Regulation of integrins is a complex interplay of structure, interacting proteins, spatial localization, protein modifications and physical forces. Many of these aspects are addressed in the review, however, the regulation by clustering and avidity changes, phosphorylation and crosstalk are completely absent, although critical for integrin function (see e.g. Bhattacharjya et al, 2022 doi.org/10.1007/s12551-022-00995-x, Gahmberg et al 2022 doi:10.1016/j.tibs.2021.11.003 and doi: 10.3390/cells11101685). These features should be included in the review.
The title implies that the review addresses T cells and LFA-1. But after reading the text it is not so clear what the focus is. A lot of work described is done with other cells than T cells and much data on integrin regulation is for other integrins, in particular beta3- integrins. Emphasis is put on the function of the intermediate form LFA-1 and how it is regulated. This is an interesting viewpoint and worth exploring, however, it may be good to include in the title. The three ways to activate LFA-1 is also described and compared, but a much more structured look describing and comparing the three signalling pathways (TCR, chemokine and selectin) and their effect on LFA-1 would make the review much more coherent and clear. A figure about selectin and chemokine signalling in migration, similar to the nice figure 2, would be helpful.
This review has commendably compiled relevant articles in the field, but with more structure, less listing of data and more synthesis this would provide an important review addressing the intermediate form of LFA-1 and LFA-1 activation downstream of selectin signalling as compared to TCR- or chemokine-mediated activation of LFA-1.
Specific points:
Page 1, row 13: The abstract states that “Recent studies show that LFA-1 is also presented in the intermediate-affinity form”. As this knowledge has been around for 20 years, I would not refer to it as recent.
Page 2, rows 63-66: Please refer to the original articles when explaining the different structural states of LFA-1, not the same review trice (Shurpf, 2001).
Page 2, row 71: The introduction mentions that LFA-1 is the only beta2-integrin expressed on T cells. This is not completely true. At least CD11c/CD18 and CD11d/CD18 are expressed on a subset of T cells, e.g. gammadelta T cells and even CD11b is found on a subset of T cells. I would rephrase it and say that LFA-1 is the predominant beta2-integrin in T cells.
Page 3, row 92: The integirn beta2 does not contain a NPxY sequence, but instead NPxF. Therefore it would be better to talk abou the commonly used NPxY/F motif. In the sentence “The talin1 head domain…” regulation for beta2 is mentioned although both articles (Vinogradova et al., 2002; Wegener et al., 2007) refer to experiments with beta3, which indeed does contain an NPxY motif. A lot of work mentioned is actually done with beta3 integrin although this article describes LFA-1 regulation. In some cases mentioning general concepts for beta-integrins can be valid, but here it would be necessary to highlight what has been proved specifically for beta2, especially since beta-integrins are regulated differently, i.e. there are no tyrosine phosphorylations on beta2. Instead, beta2 is phosphorylated on threonine and serine residues. This regulation affects complex formation and therefore LFA-1 activity. This part has been completely left out from this review and should be mentioned.
Page 3, row 128: It is unclear why there is a mention specifically of SLP-76 and its phosphorylation downstream of chemokine and selectin activation. SLP-76 is also involved in the signalling downstream of the TCR and there are several other, crucial proteins involved in the downstream signalling. Is this particularly important for LFA-1 regulation? A scheme figure of signalling downstream of the different activation modes would be helpful (as in Figure 2 for TCR signalling).
Page 3, row 133: It is mentioned that integrins link to the cytoskeleton via talin and kindlin, which may affect their activation states. The authors also claim on page 4 row 147 that it is unknown whether the intermediate form binds to the cytoskeleton. This has, however, been shown specifically for the intermediate form via alpha-actinin binding, which should be included (i.e. Stanley et al 2008, doi: 10.1038/sj.emboj.7601959, Jevnikar et al 2009 doi: 10.1002/eji.200939562).
On page 5 row 225 the reference is missing for the data on pertussin toxin’s role on leukocyte recruitment. Please add some data on how chemokines activate migration, now the text is mainly focused on selectin activation.
Parts of section 3 and 5 is mainly listing and enumerating pieces of data, the text would be improved by more synthesis and discussion.
A few sentences at the end with conclusions related to the focus of the paper seem to be missing. Now the review ends quite abruptly with the speculation about DAP12. This could include linking the mentioned diseases caused by integrin beta dysregulation to LFA-1 regulation.
